# Associations Between Problematic QQ Use and Mental Health Among Chinese Children and Adolescents: A Latent Class Analysis

**DOI:** 10.3390/brainsci15111148

**Published:** 2025-10-27

**Authors:** Li Mei, Oli Ahmed, Md Zahir Ahmed

**Affiliations:** 1School of Education, Lanzhou City University, Lanzhou 730070, China; lm@lzcu.edu.cn; 2Department of Psychology, University of Chittagong, Chattogram 4331, Bangladesh; oliahmed_polash131@cu.ac.bd; 3School of Psychology, Zhejiang Normal University, Jinhua 321004, China

**Keywords:** problematic QQ use, latent class analysis, mental health, children and adolescents, China

## Abstract

**Background:** The rise of problematic social media use among children and adolescents is often associated with significant physical and psychosocial effects. In China, QQ, a popular social media platform among youth, has become a major mental health concern due to its excessive use. The present study aimed to explore the association between QQ addiction and negative mental health through a Latent Class Analysis (LCA). **Methods:** The study data were collected from a sample of 1006 Chinese school students (49.8% male; age *M* = 13.32, *SD* = 1.34 years) through a paper-pencil survey using the convenience sampling technique. **Results:** LCA identified three latent groups based on QQ addiction symptom scores: No-risk (77.2%), At-risk (16.8%), and High-risk (6.0%). The analysis revealed that children and adolescents in the High-risk class exhibited significantly higher levels of depression, anxiety, emotional problems, conduct issues, hyperactivity, and peer problems, as well as lower life satisfaction and prosocial behaviors compared to the No-risk and At-risk groups (*p* < 0.05), signifying a strong association between problematic QQ use and poor mental health. **Conclusions:** Mental health professionals would benefit from designing intervention plans to mitigate the negative mental health among the High-risk and At-risk classes of problematic QQ users.

## 1. Introduction

With the increasing use of digital technology in the everyday lives of children and adolescents, an unprecedented rise in social media use and screen-time has been experienced worldwide [1]. Youth engagement in social media and virtual platforms has notably increased in the past decade due to the widespread connectivity, device acceptability, along with the digital shift of social and peer interactions through cultural normalization [2]. Undoubtedly, the rapid development of digital infrastructure and hyperconnectivity have paved the way for virtual connection and different social media platforms, such as WeChat, Weibo, Douyin (Chinese TikTok), and QQ, among Chinese youth compared to their global counterparts [3,4]. Among the widely used social media platforms in China, QQ is very popular with younger age groups due to its wide range of functions, like instant messaging, social networking, gaming, data sharing, and storage. With a total of 1.11 billion, China ranks first for the number of internet users in 2025 [5]; 79.7% of which are minors [6] and 562 million of which use QQ regularly [7]. From these statistics, it is signified how the young generation are immersed in virtual environments, especially on social media which often imperils mental health and well-being if used uncontrollably and for prolonged periods [8].

The benefits of the internet and social media are undeniable in many ways. For example, they foster socialization, education, and entertainment; however, they are also prone to risks if used excessively [9]. The perils of problematic social media use and addiction to social media platforms, including QQ, have gained global attention significantly due to the detrimental effects on mental health outcomes [10]. It has been well documented in diverse cultural contexts that problematic social media use is associated with negative behavioral and psychological outcomes (e.g., stress, anxiety, depression, emotional dysregulation, sleep disturbance, disrupted personal and social competence, negative academic achievement) [11,12,13,14]. This issue is also further amplified among children and adolescents who have fewer self-regulation skills and lack the coping mechanisms to balance the virtual and physical worlds effectively. As a result, this often exacerbates the negative mental health for this age group who use social media excessively [15,16].

A wide range of existing literature shows that excessive and problematic internet and social media use is associated with negative mental health, including social isolation, poor sleep quality, and disrupted life satisfaction, which elevate the risk of mental health and psychological disorders [17,18,19]. It is anticipated that this issue is more explicit in China since it has the highest number of internet users, where the minor users are significantly higher. The negative impact of problematic QQ use on mental health is an obvious concern due to the platform’s popularity and addictive nature, including excessive screen time, social withdrawal symptoms, and avoiding offline and physical activities [20]. Although a major portion of the existing literature has explored problematic internet use and internet addiction on various virtual platforms, research on the impact of problematic QQ use on the mental health of Chinese children and adolescents remains understudied.

A number of studies on internet addiction and problematic social media use adopted the variable-centered approach, which considers the addictive behavior as a continuous variable, often overlooking the heterogeneity effect of addiction within the population [21,22]. The approach often neglects the differentiation of addiction severity among the population, which may not depict the complete and diverse behavioral outcomes within the impacted groups [23]. In contrast, the person-centered approach (i.e., Latent Class Analysis [LCA]) endorses the identification of distinct subclasses within the population based on the mental health outcomes [24]. Notably, LCA has been utilized in a number of studies to analyze the heterogeneity of problematic internet use and addiction as well as its association with negative mental health outcomes [25]. However, despite these statistical advantages, this approach has not been utilized to explore problematic QQ use among Chinese children and adolescents.

### Present Study

The present study utilizes LCA to identify distinct latent subclasses of Chinese children and adolescents based on their problematic QQ use symptoms. More explicitly, this study attempts to identify the latent classes based on problematic QQ use symptoms and their differences in mental health outcomes (depression, anxiety, life satisfaction, emotional problems, conduct problems, hyperactivity, peer problems, and prosocial behaviors) across those classes. To the best of our knowledge, the present study is one of the first to utilize LCA to explore problematic QQ use and its mental health outcomes among Chinese children and adolescents through the person-centric approach rather than relying on the commonly used addiction measuring approaches. Through the distinctive identification of risk profiles, the present study also aims to provide in-depth insights into mental health implications for ensuring effective intervention plans for high-risk individuals.

## 2. Method

### 2.1. Participants

To collect the study data, a paper-pencil-based survey was conducted in 16 high schools in five cities in China (Beijing, Dongguan, Kunming, Lanzhou, and Xi’an). The minimum required sample size was calculated prior to the study, and with a statistical power of 0.90 to detect a small to moderate correlation coefficient (0.15), the minimum sample size was 463 (https://www.sample-size.net/correlation-sample-size/, accessed on 1 May 2023). Through the convenience sampling technique, 1200 survey booklets were given to the students, where the final sample included 1006 completed surveys between 30 May 2023 and 15 June 2023. A total of 194 responses were not considered due to incomplete submission. The age range of the respondents was 11 to 17 years (*M* = 13.32 years, *SD* = 1.34), with 49.8% male and 50.2% female responses. Since the study recruited minor participants, the parents or legal guardians were contacted prior to the survey to explain the nature, intention, time, cost and benefits, and data confidentiality in detail. The survey commenced once the informed consents were collected from the parents/guardians.

### 2.2. Measures

#### 2.2.1. Problematic QQ Use Scale (PQQUS)

The PQQUS consisted of 6 items to assess problematic QQ use [10]. The PQQUS assesses all six core criteria of behavioral addiction (salience, tolerance, mood modification, loss of control, withdrawal, and conflict) that have been used in the development of other social media addiction scales. Responses to the PQQUS were provided on a 5-point Likert scale ranging from 1 (very rarely) to 5 (very often). Participants responded to each item (e.g., “Spent a lot of time thinking about QQ or planned use of QQ?” and “Felt an urge to use QQ more and more?”), with the higher total score denoting higher problematic QQ use symptoms. In the present study, the PQQUS demonstrated good internal consistency with a Cronbach’s alpha of 0.857.

#### 2.2.2. Beck Anxiety Inventory (BAI)

The BAI assessed the severity of anxiety symptoms through 21 items (original version: [26]; Chinese version: [27]). Anxiety symptoms were assessed through the physiological, emotional, and cognitive components with a four-point Likert scale ranging from 0 (not at all) to 3 (severely—I could barely stand it). Participants were asked to report their feelings in the last week, including the day of the survey. Total scores of the BAI ranged from 0 to 63, with higher total scores signifying elevated anxiety levels. In the present study, the BAI demonstrated good internal consistency with a Cronbach’s alpha of 0.927.

#### 2.2.3. Beck Depression Inventory (BDI)

The BDI assessed depression severity with 21 items (original version: [28]; Chinese version: [29]). Depression symptoms were assessed through the physiological, emotional, and cognitive components with a four-point Likert scale ranging from 0 (not at all) to 3 (severely—I could barely stand it). Total scores of the BDI ranged from 0 to 63, with higher total scores signifying elevated depression levels. In the present study, the BDI demonstrated good internal consistency with a Cronbach’s alpha of 0.907.

#### 2.2.4. Satisfaction with Life Scale (SWLS)

The SWLS is a five-item scale (original version: [30]; Chinese version: [31]) and was used to assess individual satisfaction levels with one’s global life from a cognitive judgmental perspective. Using a seven-point Likert scale ranging from 1 (strongly disagree) to 7 (strongly agree), the respondents rated their satisfaction level with life (sample items: “In most ways my life is close to my ideal,” “The conditions of my life are excellent”). The SWLS had a total score ranging from 5 to 35, with the higher scores signifying higher levels of life satisfaction. In the present study, the SWLS demonstrated good internal consistency with a Cronbach’s alpha of 0.883.

#### 2.2.5. The Strengths and Difficulties Questionnaire (SDQ)

The SDQ assessed the behavioral problems for youth using 25 items (original version: [32]; Chinese version: [33]). Respondents rated their behavior over the last six months on the five subscales: emotional problems, conduct problems, hyperactivity/inattention, peer relationship problems, and prosocial behaviors. Items were rated on a 3-point Likert scale (“Not true,” “Somewhat true,” or “Certainly true”). The total score ranged from 0 to 10 for each subscale, with higher total scores signifying more problems, with the exception of the prosocial subscale. In the present study, the overall SDQ demonstrated good internal consistency with a Cronbach’s alpha of 0.774 (emotional problems: 0.694, conduct problems: 0.677, hyperactivity: 0.675, peer problems: 0.709, prosocial behaviors: 0.698).

### 2.3. Statistical Analysis

LCA was conducted based on problematic QQ use symptoms to identify the distinct latent subclasses. The test was run from two- to five-class solutions. Several fit indices, including the Akaike Information Criterion (AIC), the Bayesian Information Criterion (BIC), the sample-size-adjusted Bayesian Information Criterion (SABIC), entropy, class size, and the Lo-Mendell-Rubin adjusted likelihood ratio test (LMRT), were assessed to determine the number of latent classes. These indices help to determine the model fit to identify the best solution with the explanatory power. A one-way analysis of variance (ANOVA) was conducted subsequently to assess the differences in mental health outcomes among latent classes (No-risk, At-risk, and High-risk). Additionally, a post hoc (i.e., Fisher’s Least Significance Difference) analysis was conducted to determine the significant differences among the classes based on the study variables.

## 3. Results

### 3.1. Latent Class Analysis Results

Latent Class Analysis results are presented in Table 1. Fit indices (AIC, BIC, SABIC, entropy, LMR test value and its *p*-value, class size, and average class probabilities for most likely latent class membership by latent class) were determined from a two-class solution to a five-class solution. Table 1 shows that the five-class solution had the lowest AIC (11,040.764), BIC (11,237.314), and SABIC (11,110.271) values compared to the other solutions. These fit indices suggested five latent classes. The entropy value of the five-class solution was highest (0.969), suggesting better classification, although all other solutions had the accepted range of entropy values (>0.80). However, LMRT values did not support both four-class and five-class solutions as they were non-significant (>0.05), suggesting three latent classes with a reasonably fit entropy value of 0.950 and a significant LMRT value *(p* = 0.003). The values of the class size additionally rejected four-class and five-class solutions as both classes had observations lower than 5% (four-class solution: 4.7%; five-class solution: 3.6%). Average class probabilities for most likely latent class membership by latent class also suggested that the three-class solution demonstrated comparatively higher average probabilities for latent class membership (0.986, 0.935, and 0.993, respectively). Based on these fit statistics, the three-class solution (three latent classes) was considered for this study.

### 3.2. Descriptive Statistics of Problematic QQ Use Symptoms

Table 2 shows the descriptive statistics (means and standard deviations) for the problematic QQ use symptoms and total QQ addiction score. Considering the classification by the risk level, the No-risk class demonstrated comparatively lower scores for all the variables than the At-risk and High-risk classes. Additionally, the High-risk class demonstrated the highest scores for all problematic QQ use symptoms and total QQ addiction scores. Based on the symptoms and addiction scores, the three classes were labeled veritably as No-risk, At-risk, and High-risk (Figure 1).

Spearman’s rho correlation results among the variables are presented in Table 3. The correlation analysis highlighted significant associations between problematic QQ use and key study variables. The findings support that problematic QQ use is associated with negative mental health, but no significant correlation was found with prosocial behavior.

### 3.3. Mean Differences in Latent Classes

Table 4 shows the one-way ANOVA findings (mean differences in the latent classes for depression, anxiety, life satisfaction, emotional problems, conduct problems, hyperactivity, peer problems, and prosocial behavior). Results demonstrated different significance levels among No-risk, At-risk, and High-risk classes for the key study variables. Depression (*F* = 45.730, *p* < 0.001, ω^2^ = 0.082) and anxiety (*F* = 34.792, *p* < 0.001, ω^2^ = 0.063) demonstrated medium effects, suggesting a moderate to strong impact of problematic QQ use. Life satisfaction (*F* = 19.965, *p* = 0.001, ω^2^ = 0.036) and emotional problems (*F* = 19.885, *p* = 0.001, ω^2^ = 0.036) have small to medium effects; and conduct problems (*F* = 14.058, *p* < 0.001, ω^2^ = 0.025), hyperactivity (*F* = 15.468, *p* < 0.001, ω^2^ = 0.028), and peer relationship problems (*F* = 8.146, *p* = 0.001, ω^2^ = 0.014) demonstrated small effects. Additionally, prosocial behaviors demonstrated a very small effect (*F* = 4.072, *p* = 0.017, ω^2^ = 0.006), indicating the least effect with problematic QQ use.

### 3.4. Post Hoc Test Results

Table 5 summarized the post hoc results of the one-way ANOVA. For the score of depression, the At-risk class was significantly higher than the No-risk class (mean difference = −4.70, *p* < 0.001, 95% CI [−6.13, −3.28]), but the High-risk class scored significantly higher than the At-risk class (mean difference = −4.15, *p* < 0.001, 95% CI [−6.68, −1.63]). Compared to the No-risk class, the High-risk class demonstrated the highest depression score (mean difference = −8.86, *p* < 0.001, 95% CI [−11.11, −6.61]). For anxiety, a similar trend was observed when comparing all the classes, where the High-risk class scored significantly higher than the At-risk class (mean difference = −2.38, *p* = 0.029, 95% CI [−4.53, −0.24]) and the No-risk class (mean difference = −6.16, *p* < 0.001, 95% CI [−8.08, −4.25]). Similarly, the At-risk class also demonstrated a higher anxiety score than the No-risk class (mean difference = −3.78, *p* < 0.001, 95% CI [−8.08, −4.25]).

As anticipated, the scores for life satisfaction demonstrated opposite trends, where the No-risk class scored significantly higher satisfaction than both the At-risk (mean difference = 3.22, *p* < 0.001, 95% CI [2.12, 4.33]) and High-risk classes (mean difference = 2.95, *p* < 0.001, 95% CI [1.20, 4.69]). The class difference between the At-risk and High-risk classes was not significant (mean difference = −0.28, *p* = 0.783, 95% CI [−2.23, 1.68]). For emotional problems, the High-risk class was higher than the No-risk class (mean difference = −1.63, *p* < 0.001, 95% CI [−2.25, −1.01]) and the At-risk class (mean difference = −0.78, *p* = 0.029, 95% CI [−1.48, −0.08]). A similar pattern was also observed regarding conduct problems, where the No-risk class demonstrated a significantly lower score than both the At-risk (mean difference = −0.86, *p* < 0.001, 95% CI [−1.22, −0.49]) and High-risk classes (mean difference = −0.89, *p* < 0.001, 95% CI [−1.46, −0.32]). However, the difference between the At-risk and High-risk classes was non-significant for conduct problems (mean difference = −0.04, *p* = 0.911, 95% CI [−0.68, −0.60]). Regarding hyperactivity, the No-risk class scored significantly lower than the At-risk (mean difference = −0.83, *p* < 0.001, 95% CI [−1.21, −0.45]) and High-risk classes (mean difference = −1.25, *p* < 0.001, 95% CI [−1.86, −0.65]). However, no significant difference was observed for hyperactivity between the At-risk and High-risk classes (mean difference = −0.42, *p* = 0.223, 95% CI [−1.10, 0.26]). For peer problems, the No-risk class had a significant difference from the At-risk (mean difference = −0.57, *p* = 0.003, 95% CI [−0.94, −0.19]) and High-risk classes (mean difference = −0.92, *p* = 0.002, 95% CI [−1.51, −0.33]). However, there was a non-significant difference for peer problems between the At-risk and High-risk classes (mean difference = −0.35, *p* = 0.304, 95% CI [−1.10, −0.32]). Lastly, for prosocial behavior, the At-risk class showed lower scores than the No-risk class (mean difference = −0.57, *p* = 0.005, 95% CI [0.17, 0.97]); and the High-risk class did not significantly differ from the No-risk and At-risk classes.

### 3.5. Regression Analysis of the Impact of Problematic QQ Use on Mental Health

Table 6 presents the regression analysis results comparing the impact of problematic QQ use on mental health outcomes. For depression, both the comparison between the No-risk class and the At-risk class (β = 0.028, *p* = 0.024) and the No-risk class and the High-risk class (β = 0.061, *p* < 0.001) showed significant positive relationships. This suggests that higher problematic QQ use is associated with elevated depression symptoms. For anxiety scores, a similar trend was observed for the comparison between the No-risk class and the At-risk class (β = 0.0345, *p* = 0.008); however, the comparison between the No-risk class and the High-risk class was not significant (β = 0.033, *p* = 0.070). Regarding the results of life satisfaction, the No-risk class had a significantly higher level of life satisfaction compared to the At-risk class (β = −0.038, *p* = 0.004); however, the comparison between the No-risk class and the High-risk class was found to be non-significant (β = −0.015, *p* = 0.490). This suggests that problematic QQ use significantly impacts life satisfaction among the At-risk class. No significant relationships were found for other mental health outcomes (i.e., emotional problems, conduct problems, hyperactivity, peer problems, and prosocial behaviors) among the classes. Furthermore, moderation was conducted for sex, but there were no significant moderation effects on the associations between latent groups of QQ users and mental health outcomes (see Appendix A).

## 4. Discussion

This cross-sectional study examined the associations between problematic QQ use and different mental health symptomatology among Chinese children and adolescents using Latent Class Analysis (LCA). The findings of the study highlighted a significant association between problematic QQ use and QQ addiction with mental health outcomes: depression, anxiety, emotional and conduct problems, hyperactivity, peer problems, and reduced life satisfaction. Comprehensive LCA identified three distinct classes of problematic QQ users: No-risk (77.2%), At-risk (16.8%), and High-risk (6.0%); with the comparatively higher-risk groups demonstrating elevated addiction symptoms and negative mental health. To date, this is the first study that has explored the latent groups of QQ users based on problematic QQ use symptoms. The findings are consistent with other studies exploring the latent groups of users based on other social media sites, including Facebook and overall social media usage [34,35,36,37]. These findings align with existing literature on behavioral addiction and its adverse mental health impacts, highlighting the importance of early diagnosis and tailored intervention plans for individuals categorized in risk groups for developing behavioral addiction.

The findings derived from the LCA highlighted comorbidity of all six components of addiction, along with higher negative mental health (depression, anxiety, life satisfaction, emotional problems, conduct problems, hyperactivity, and peer problems), with significant mean differences found. However, the differences in mental health symptomatology were not the same across the groups. Larger differences existed with depression and anxiety, while small to moderate differences were found with life satisfaction and emotional problems. Small differences were found with conduct problems, hyperactivity, and peer problems. Although differences were varied, these overall suggested vulnerability of the high-risk group of QQ users.

Findings further showed significant associations between latent groups of QQ users and mental health symptomatology. The High-risk group had a significant relationship with depression compared to the No-risk group, which is in line with previous studies showing the association between excessive internet use and depressive symptoms among adolescents [38,39,40,41,42]. It is highly anticipated that problematic QQ use contributes to elevated depressive symptoms through inducing social isolation, disrupted sleep quality, and confining other offline activities (e.g., fostering hobbies and outdoor recreational activities), which all ultimately result in negative mental health. Similar to depression, At-risk group had a significant association with a higher level of anxiety compared to the No-risk group. This finding suggested different levels of mental health vulnerability across the groups of QQ users. This association prevails among adolescents whose primary source of socialization is based on virtual presence [43,44,45,46,47,48]. To explain this association, several contributing factors may be of relevance, such as the urge to maintain online presence, fear of missing out, and social desirability. Often, the validation-seeking behaviors (for example: likes, shares, comments, and other interactions) lead to emotional distress, where adolescents can be more vulnerable to mental health consequences [49,50,51,52]. Cultural context plays a significant role in shaping the addictive behaviors of Chinese adolescents with regard to QQ use. Parenting style and academic stressors often play key roles in the development and prevention of problematic internet use in China [53]. In China, QQ is used not only to cope with academic stress but also for socialization, which can contribute to excessive and problematic use. Additionally, the multifaceted use of QQ (e.g., entertainment, social interactions, academic activities) makes it highly popular among adolescents.

Our findings also demonstrated a significant relationship between At-risk users and significantly lower levels of life satisfaction among the High-risk class compared to the No-risk users. This result is also supported by previous studies that highlight how problematic internet use has multifaceted negative impacts on life satisfaction [54,55,56,57]. One of the plausible reasons for the lower level of life satisfaction is due to excessive QQ use, which can reduce real-life physical activities and relationships. Adolescents who prioritize their virtual presence over physical interactions often also feel lonely and isolated, which can jeopardize their overall well-being [58,59,60].

Considering the behavioral problems, the High-risk group was more prevalent than the other two latent groups. The High-risk group reported significantly elevated levels of emotional problems, conduct problems, hyperactivity, and peer problems compared to the other two classes. Our finding concerning behavioral externalization was consistent with the findings of previous research, where problematic internet use was found to be positively associated with poor social skills, aggressive behavior, and impulsive behavior [61,62,63,64]. The best possible explanation for the higher conduct and emotional problems among the High-risk class may be due to the addictive nature of problematic QQ use, which can result in excessive irritability, impaired emotional regulation, and disrupted interpersonal relationships, which also has backup from the literature [11,65,66,67]. Additionally, the higher levels of peer problems among the High-risk group may be explained through the lens of social isolation and loneliness that are often very common among excessive internet users. Adolescents from the High-risk group in our study may feel lonely due to their excessive internet use, resulting in disrupted peer relationships.

In contrast to the significant differences among the study variables, only prosocial behavior did not demonstrate any significant differences among groups, with slightly lower, hence non-significant, levels of prosocial behaviors among the High-risk group compared to the No-risk group. This finding indicates that problematic QQ use is associated with negative mental health and behavioral outcomes; interestingly, prosocial behavior did not demonstrate statistical significance. This contradicts the existing literature [68,69,70] and may be due to maintaining physical or offline connections along with excessive virtual presence. Future longitudinal research could possibly explain the complex underlying mechanism.

Although the present study failed to explore any significant associations between latent groups of QQ users with emotional and conduct problems hyperactivity, peer problems, and prosocial behaviors, the significant mean differences found among mental health symptomatology do suggest possible vulnerability among At-risk and High-risk users. However, longitudinal and experimental studies (identifying At-risk and High-risk users, introducing interventions to reduce problematic usage symptoms, and examining the results) are needed to confirm these associations. Another important finding of the present study is related to the effect of sex on the associations between latent groups of QQ users and mental health. Although existing literature suggests that females are more susceptible to negative mental health outcomes from excessive internet use [43,44], no moderating effect of sex on the key study variables was found in this study. This suggests that sex did not significantly influence the development of negative mental health due to problematic QQ use.

## 5. Limitations and Future Research Implications

The present study acknowledges some limitations. First, self-report data are often prone to social desirability and recall bias. Respondents can often answer in a favorable state that distorts the information, which may lead to inaccurate findings. It is anticipated that participants underreported their problematic QQ use and overlooked the negative mental health and symptoms. Further longitudinal research could minimize those biases through objective measures, such as screen-time tracking and clinical diagnosis of mental health symptoms. Second, cross-sectional studies have the limitation of establishing causal relationships between the study variables. Although the association between problematic QQ use and negative mental health was highlighted, it remains unclear whether this is due to problematic QQ use or any other pre-existing conditions that lead to the problematic use. Future longitudinal studies could reveal the relationships over time with proper directionality. Third, recruitment of the participants using a convenience sampling technique raises a question about the generalizability of this study. A wide and diversified representation of the data will help to address this issue in future studies. Fourth, the lower Cronbach’s alpha in some of the subscales (emotional problems, conduct problems, hyperactivity, and prosocial behaviors) of the SDQ may limit the generalizability of the findings. Fifth, this study did not highlight the underlying motivational mechanisms (e.g., stress, loneliness, coping mechanisms) of problematic QQ use. Future research should explore the driving factors and underlying motivation for excessive social media usage. Finally, future longitudinal studies should highlight the association between problematic QQ use and other behavioral and digital addictions to reveal the complex negative effects on adolescents’ mental health and well-being.

## 6. Conclusions

The present study explored the association between problematic QQ use and mental health outcomes among Chinese adolescents through latent classification. The study demonstrated that the High-risk class reported significantly higher depression, anxiety, emotional and conduct problems, along with lower levels of life satisfaction compared to the No-risk and At-risk classes. It is necessary to consider the unique cultural and social impact of problematic QQ use in China, where parental style, academic stress, and social interaction are important influences. These findings affirm the importance of early diagnosis and intervention for children and adolescents who are at risk of problematic QQ use, especially those classified as At-risk and High-risk individuals. The findings of the present study highlight the mental health tolls of problematic QQ use; as a result, mental health professionals and educators are encouraged to develop reasonable intervention strategies emphasizing the role of personality and individual differences of minors in digital settings. Effective intervention programs should aim to consider cultural norms to promote healthier digital presence among adolescents.

## Figures and Tables

**Figure 1 brainsci-15-01148-f001:**
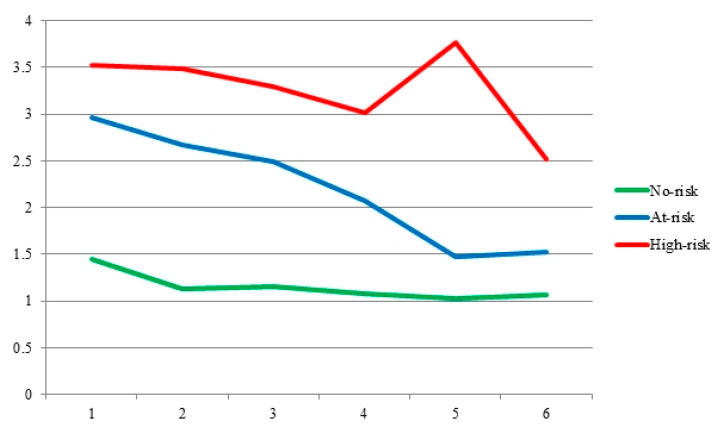
Mean comparison of the three identified latent profiles.

**Table 1 brainsci-15-01148-t001:** Fit indices, class size and average class probabilities for most likely latent class membership by latent class.

Solutions	AIC	BIC	SABIC	Entropy	LMRT (*p* Value)	Class Size	Average Class Probabilities for Most Likely Latent Class Membership by Latent Class				
							**1**	**2**	**3**	**4**	**5**
2	12,783.848	12,877.209	12,816.864	0.967	2408.621 (<0.0179)	886 (88.1%)	0.993	0.007			
						120 (11.9%)	0.030	0.970			
**3**	**11,860.029**	**11,987.786**	**11,905.209**	**0.950**	**918.833 (0.0030)**	**777 (77.2%)**	**0.986**	**0.014**	**0.000**		
						**169 (16.8%)**	**0.060**	**0.935**	**0.005**		
						**60 (6.0%)**	**0.000**	**0.007**	**0.993**		
4	11,429.610	11,591.764	11,486.953	0.959	435.422 (0.2449)	740 (73.6%)	0.987	0.000	0.013	0.000	
						47 (4.7%)	0.001	0.994	0.004	0.001	
						162 (16.1%)	0.064	0.003	0.933	0.001	
						57 (5.7%)	0.000	0.023	0.006	0.972	
5	11,040.764	11,237.314	11,110.271	0.969	435.279 (0.6993)	738 (73.4%)	0.986	0.013	0.000	0.000	0.000
						93 (9.2%)	0.078	0.822	0.000	0.000	0.000
						103 (10.2%)	0.011	0.004	0.979	0.000	0.006
						36 (3.6%)	0.000	0.000	0.004	0.996	0.000
						36 (3.6%)	0.003	0.002	0.000	0.000	0.994

*Note.* AIC = Akaike information criterion; BIC = Bayesian information criterion; SABIC = sample-size-adjusted Bayesian information criterion; LMRT = Lo-Mendell-Rubin adjusted likelihood ratio test.

**Table 2 brainsci-15-01148-t002:** Descriptive statistics (mean and standard deviations) of problematic QQ use symptoms, and total QQ addiction score.

Variable	Overall		No Risk		At Risk		High Risk	
	*M*	*SD*	*M*	*SD*	*M*	*SD*	*M*	*SD*
Salience	1.83	1.05	1.45	0.68	2.96	1.03	3.52	1.03
Tolerance	1.53	0.95	1.13	0.37	2.67	0.90	3.48	1.28
Mood modification	1.51	0.96	1.16	0.44	2.49	1.14	3.30	1.37
Relapse	1.36	0.84	1.08	0.39	2.07	1.08	3.02	1.20
Withdrawal	1.27	0.74	1.03	0.20	1.47	0.60	3.77	0.81
Conflict	1.24	0.64	1.07	0.34	1.53	0.76	2.52	1.26
Addiction score	8.74	4.01	6.94	1.31	13.20	2.39	19.60	3.93

**Table 3 brainsci-15-01148-t003:** Spearman’s Rho Correlations Among the Variables.

Variable	1	2	3	4	5	6	7	8	9
1. Problematic QQ Use	1.000	0.229 **	0.230 **	−0.190 **	0.182 **	0.178 **	0.178 **	0.129 **	−0.069
2. Depression	0.229 **	1.000	0.605 **	−0.375 **	0.441 **	0.281**	0.329 **	0.227 **	−0.182 **
3. Anxiety	0.230 **	0.605 **	1.000	−0.297 **	0.418 **	0.167 **	0.269 **	0.217 **	−0.090
4. Life satisfaction	−0.190 **	−0.375 **	−0.297 **	1.000	−0.218 **	−0.201 **	−0.254 **	−0.194 **	0.171 **
5. Emotional problems	0.182 **	0.441 **	0.418 **	−0.218 **	1.000	0.450 **	0.453 **	0.447 **	−0.114 **
6. Conduct problems	0.178 **	0.281 **	0.167 **	−0.201 **	0.450 **	1.000	0.461 **	0.597 **	−0.333 **
7. Hyperactivity	0.178 **	0.329 **	0.269 **	−0.254 **	0.453 **	0.461 **	1.000	0.380 **	−0.316 **
8. Peer problems	0.129 **	0.227 **	0.217 **	−0.194 **	0.447 **	0.597 **	0.380 **	1.000	−0.172 **
9. Prosocial behaviors	−0.069	−0.182 **	−0.090	0.171 **	−0.114 **	−0.333 **	−0.316 **	−0.172 **	1.000

*Note.*** *p* < 0.01.

**Table 4 brainsci-15-01148-t004:** Mean differences in latent classes in depression, anxiety, life satisfaction, emotional problems, conduct problems, hyperactivity, peer problems, and prosocial behaviors.

	No Risk *M (SD)*	At Risk *M (SD)*	High Risk *M (SD)*	*df*	*F*-Value	Sig.	ω^2^
Depression	7.12 (7.49)	11.83 (10.28)	15.98 (14.39)	2, 1003	45.730	<0.001	0.082
Anxiety	24.55 (6.37)	28.33 (9.48)	30.72 (10.49)		34.792	<0.001	0.063
Life satisfaction	29.18 (6.37)	25.96 (7.32)	26.23 (8.00)		19.965	<0.001	0.036
Emotional problems	2.52 (2.30)	3.37 (2.53)	4.15 (2.72)		19.885	<0.001	0.036
Conduct problems	2.46 (2.16)	3.31 (2.24)	3.35 (2.18)		14.058	<0.001	0.025
Hyperactivity	2.89 (2.29)	3.73 (2.30)	4.15 (2.43)		15.468	<0.001	0.028
Peer problems	3.10 (2.24)	3.67 (2.21)	4.02 (2.44)		8.146	<0.001	0.014
Prosocial behaviors	7.27 (2.40)	6.69 (2.43)	7.32 (2.41)		4.072	0.017	0.006

*Note. M* = mean, *SD* = standard deviation, Sig. = Significance, *df* = Degree of Freedom, ω^2^ = Omega-Squared Effect Size.

**Table 5 brainsci-15-01148-t005:** Post hoc test results among latent profiles in depression, anxiety, life satisfaction, emotional problems, conduct problems, hyperactivity, peer problems, and prosocial behaviors.

Variable	(I)	(J)	Mean Difference (I-J)	Sig.	95% Confidence Interval		Cohen’s *d*
					**Lower Bound**	**Upper Bound**	
Depression	No-risk	At-risk	−4.70	<0.001	−6.13	−3.28	0.55
		High-risk	−8.86	<0.001	−11.11	−6.61	1.04
	At-risk	High-risk	−4.15	<0.001	−6.68	−1.63	0.49
Anxiety	No-risk	At-risk	−3.78	<0.001	−4.99	−2.57	0.52
		High-risk	−6.16	<0.001	−8.08	−4.25	0.85
	At-risk	High-risk	−2.38	0.029	−4.53	−0.24	0.33
Life satisfaction	No-risk	At-risk	3.22	<0.001	2.12	4.33	0.49
		High-risk	2.95	<0.001	1.20	4.69	0.44
	At-risk	High-risk	−0.28	0.783	−2.23	1.68	0.04
Emotional problems	No-risk	At-risk	−0.85	<0.001	−1.24	−0.46	0.36
		High-risk	−1.63	<0.001	−2.25	−1.01	0.69
	At-risk	High-risk	−0.78	0.029	−1.48	−0.08	0.33
Conduct problems	No-risk	At-risk	−0.86	<0.001	−1.22	−0.49	0.40
		High-risk	−0.89	0.002	−1.46	−0.32	0.41
	At-risk	High-risk	−0.04	0.911	−0.68	0.60	0.02
Hyperactivity	No-risk	At-risk	−0.83	<0.001	−1.21	−0.45	0.36
		High-risk	−1.25	<0.001	−1.86	−0.6459	0.54
	At-risk	High-risk	−0.42	0.223	−1.10	0.26	0.18
Peer problems	No-risk	At-risk	−0.57	0.003	−0.94	−0.19	0.25
		High-risk	−0.92	0.002	−1.51	−0.33	0.41
	At-risk	High-risk	−0.35	0.304	−1.01	0.32	0.16
Prosocial behaviors	No-risk	At-risk	0.57	0.005	0.17	0.97	0.24
		High-risk	−0.05	0.876	−0.68	0.58	0.02
	At-risk	High-risk	−0.62	0.084	1.33	0.08	0.26

**Table 6 brainsci-15-01148-t006:** Regression analysis of the impact of problematic QQ use on mental health.

Predictor	B	*SE*	β	*p*-Value	95% CI	
					Lower	Upper
**Depression** (Model Summary: *F* = 21.924, *df* (5, 1000), *p* < 0.001, *R*^2^ = 0.314)						
No-risk vs. At-risk	1.028	0.012	0.028	0.024	1.004	1.054
No-risk vs. High-risk	1.063	0.017	0.061	<0.001	1.027	1.099
**Anxiety** (Model Summary: *F* = 16.163, *df* (5, 1000), *p* < 0.001, *R*^2^ = 0.273)						
No-risk vs. At-risk	1.034	0.013	0.034	0.008	1.009	1.060
No-risk vs. High-risk	1.034	0.018	0.033	0.070	0.997	1.071
**Life satisfaction** (Model Summary: *F* = 8.250, *df* (5, 1000), *p* < 0.001, *R*^2^ = 0.199)						
No-risk vs. At-risk	0.963	0.013	−0.038	0.004	0.939	0.988
No-risk vs. High-risk	0.986	0.021	−0.015	0.490	0.946	1.027
**Emotional problems** (Model Summary: *F* = 10.292, *df* (5, 1000), *p* < 0.001, *R*^2^ = 0.199)						
No-risk vs. At-risk	0.988	0.048	0.012	0.797	0.899	1.085
No-risk vs. High-risk	1.025	0.075	0.025	0.740	0.885	1.187
**Conduct problem** (Model Summary: *F* = 6.937, *df* (5, 1000), *p* < 0.001, *R*^2^= 0.183)						
No-risk vs. At-risk	1.088	0.052	0.084	0.103	0.983	1.204
No-risk vs. High-risk	1.025	0.083	0.025	0.762	0.872	1.206
**Hyperactivity** (Model Summary: *F* = 7.282, *df* (5, 1000), *p* < 0.001, *R*^2^ = 0.187)						
No-risk vs. At-risk	1.029	0.046	0.029	0.528	0.941	1.126
No-risk vs. High-risk	1.086	0.069	0.083	0.228	0.950	1.243
**Peer problems** (Model Summary: *F* = 3.868, *df* (5, 1000), *p* = 0.002, *R*^2^ = 0.138)						
No-risk vs. At-risk	0.991	0.048	−0.009	0.856	0.902	1.090
No-risk vs. High-risk	1.034	0.075	0.034	0.655	0.892	1.199
**Prosocial behaviors** (Model Summary: *F* = 3.598, *df* (5, 1000), *p* = 0.003, *R*^2^ = 0.133)						
No-risk vs. At-risk	0.971	0.039	−0.029	0.454	0.899	1.049
No-risk vs. High-risk	1.110	0.067	0.105	0.117	0.974	1.266

*Note.* Cox and Snell R^2^ = 0.128; Nagelkerke R^2^ = 0.174.

## Data Availability

The data presented in this study are available on request from the corresponding author due to confidentiality restrictions.

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
