# Peer review of "Associations Between Problematic QQ Use and Mental Health Among Chinese Children and Adolescents: A Latent Class Analysis"

_brainsci, 2025, doi:10.3390/brainsci15111148_

Round 1
Reviewer 1 Report
Comments and Suggestions for Authors
I want to thank the Authors and the Editors for the opportunity to review the article submitted to MDPI’s Brain Sciences. The reviewed manuscript refers to the use of LCA while testing the relationship between problematic social media usage and mental health among adolescents. The presented article is of good quality. The introduction presents the manuscript's subject matter concisely yet thoroughly, and the discussion attempts to analyse the obtained results. Below you will find my comments on the individual sections of the authors’ manuscript, which can benefit from additional revisions.
Participants: Please specify if the number of 1200 surveys was calculated a priori with the use of the statistical power analysis or was chosen arbitrarily?
Measures: The PQQUS scale uses 6 criteria of behavioural addiction, widely described by Mark Griffiths, and was developed by two authors of the reviewed manuscript. Please explain why the authors have decided to use a scale specifically designed for QQ in mind, instead of another, more universal scale that can test problematic social media use? On the one hand, using a scale that has a specific platform in mind has huge advantages, but limits the possible inclusion in meta-analyses. The PQQUS scale does not use items that refer to any of the QQ’s specific features, so if we remove the “QQ” from its items, it can be used in reference to any other social media platform. Please elaborate on that matter.
Measures: In the description of the BAI, BDI, and SWLS scales, we can find the information about their psychometric properties, such as CFI=1.00 and RMSEA=0.00. It might suggest that the tested model was over-specified (saturated) and that too many covariance errors were established in order to gain a suitable model fit, or their measurement was not corrected based on the size of the sample (N>1000). Please elaborate on that. I highly encourage the authors to post their full results report as supplementary material or in any open-access archive, such as OSF.
Measures: Why did the authors decide to report model fit measures based on their own sample for BAI, BDI, SWLS, SDQ, but not PQQUS? I encourage the authors to make their scale descriptions consistent.
Statistical analysis (section 2.3): Please state if any threshold values were chosen for BIC and AIC model fit indices a priori to the statistical analysis.
Results: I highly recommend that the authors supplement this section with the information on the relationship between the tested variables, with approaches such as the network analysis, or at least a Pearson’s r correlation, especially since in section 3.5 they test the one-sided relationships with the use of the regression analysis. If the authors plan to publish an additional article on such relationships, please state that in the manuscript to justify the possible “salami-slicing” allegations in the future.
Results: in Table 3, please add the information on the degrees of freedom (df), which can be helpful to the reader in verifying if the results are free of any p-hacking practices.
Results: Additionally, I highly recommend that the authors calculate the omega-squared effect size measure, which is less biased (the authors can refer to multiple publications by Daniel Lakens on that matter).
Results: in Table 3, eight effect size measure values are reported. Out of those 8 partial eta-squared values, 2 can be interpreted as small, 3 can be interpreted as moderate, and only 2 can be interpreted as large (above .05). The authors do not discuss the effect size values in both the Results section as well as the Discussion. It’s crucial to describe both the effect size measures and significance coefficients (such as p-values and confidence intervals). There is a difference between saying that two groups are different and stating that those groups are significantly different, but the size of the difference is very small.
Results: I believe that Table 4, representing the post-hoc analysis, should also report the effect size measure, such as Cohen’s d values.
Results: There is a small error in Table 5, since there is no such thing as an unstandardised beta coefficient. Did the authors mean to report the non-standardized OLS regression coefficient? Unstandardized regression coefficient should be reported as B (a capital letter b), while the standardized one is a Beta (represented by the Greek letter β).
Results: I believe that the authors should report model specifications for the regression model: F-value, degrees of freedom, p-value, next to the R-squared values under the table.
Discussion: I highly encourage the authors to supplement this section with the size of the effect size measures in mind, as mentioned in my previous comment.
Author Response
I want to thank the Authors and the Editors for the opportunity to review the article submitted to MDPI’s Brain Sciences. The reviewed manuscript refers to the use of LCA while testing the relationship between problematic social media usage and mental health among adolescents. The presented article is of good quality. The introduction presents the manuscript's subject matter concisely yet thoroughly, and the discussion attempts to analyse the obtained results. Below you will find my comments on the individual sections of the authors’ manuscript, which can benefit from additional revisions.
Comment 1# Participants: Please specify if the number of 1200 surveys was calculated a priori with the use of the statistical power analysis or was chosen arbitrarily?
Author’s Response# Thank you for addressing this issue. In response to that, we calculated the sample size prior to study and with a statistical power of 90% to detect the small to moderate correlation coefficient (0.15), the minimum sample size was 463. We recruited a total of 1200 participants, which was far more than the expected number. For more clarity we added the following in the Participants subsection:
“The minimum required sample size was calculated prior to the study, and with a statistical power of 0.90 to detect a small to moderate correlation coefficient (0.15), the minimum sample size was 463 (https://www.sample-size.net/correlation-sample-size/).”
Comment 2# Measures: The PQQUS scale uses 6 criteria of behavioural addiction, widely described by Mark Griffiths, and was developed by two authors of the reviewed manuscript. Please explain why the authors have decided to use a scale specifically designed for QQ in mind, instead of another, more universal scale that can test problematic social media use? On the one hand, using a scale that has a specific platform in mind has huge advantages, but limits the possible inclusion in meta-analyses. The PQQUS scale does not use items that refer to any of the QQ’s specific features, so if we remove the “QQ” from its items, it can be used in reference to any other social media platform. Please elaborate on that matter.
Author’s Response# Thank you for your valuable comment. We administrated the PQQUS scale which was developed to assess the problematic QQ use effectively considering all the key behavioral addiction criteria proposed by Griffiths (2005). The scale is highly relevant to the Chinese youth and the scale distinctively validated to use in this specific context. Most importantly, we recognize there might be arise of potential measurement bias if we uses general problematic social media use scale in this context.
About QQ feature: Similar to the base Bergen Social Media Addiction Scale (Andreassen et al., 2016), we kept our focus on QQ addiction symptoms rather than specific features of this social media platform. Therefore, we did not referred to any specific features of QQ.
Comment 3# Measures: In the description of the BAI, BDI, and SWLS scales, we can find the information about their psychometric properties, such as CFI=1.00 and RMSEA=0.00. It might suggest that the tested model was over-specified (saturated) and that too many covariance errors were established in order to gain a suitable model fit, or their measurement was not corrected based on the size of the sample (N>1000). Please elaborate on that. I highly encourage the authors to post their full results report as supplementary material or in any open-access archive, such as OSF.
Author’s Response# We sincerely apologize for this issue. We intended to report Cronbach's alpha for each scale and mistakenly reported the model fit which was not for these scales. We have now removed the scale’s model from the revised manuscript.
Comment 4# Measures: Why did the authors decide to report model fit measures based on their own sample for BAI, BDI, SWLS, SDQ, but not PQQUS? I encourage the authors to make their scale descriptions consistent.
Author’s Response# Thank you for pointing out this inconsistency, we now reported the Cronbach's alpha for all the scales.
In this present study, the PQQUS demonstrated good internal consistency with a Cronbach’s alpha of 0.857.
In this present study, the BAI demonstrated good internal consistency with a Cronbach’s alpha of 0.927. .
In this present study, the BDI demonstrated good internal consistency with a Cronbach’s alpha of 0.907.
In this present study, the SWLS demonstrated good internal consistency with a Cronbach’s alpha of 0.883.
In this present study, the overall SDQ demonstrated good internal consistency with a Cronbach’s alpha of 0.774 (emotional problems: 0.694, conduct problem: 0.677, hyperactivity: 0.675, peer problems: 0.709, prosocial behaviors: 0.698).
Comment 5# Statistical analysis (section 2.3): Please state if any threshold values were chosen for BIC and AIC model fit indices a priori to the statistical analysis.
Author’s Response# There are no specific threshold values for BIC and AIC. Smaller BIC and AIC values suggest better model fits. However, the better fitted model in latent class analysis is determined by AIC, BIC, entropy, number of observations in each latent groups, theoretical relevance. We prioritized all of these points to identify the better fitted model.
Comment 6#: Results: I highly recommend that the authors supplement this section with the information on the relationship between the tested variables, with approaches such as the network analysis, or at least a Pearson’s r correlation, especially since in section 3.5 they test the one-sided relationships with the use of the regression analysis. If the authors plan to publish an additional article on such relationships, please state that in the manuscript to justify the possible “salami-slicing” allegations in the future.
Author’s Response# Thank you for your valuable suggestion. We added Spearman’s Rho correlation analysis results in the Results section (Table 3) which showed the relationships between the key variables.
Comment 7# Results: in Table 3, please add the information on the degrees of freedom (df), which can be helpful to the reader in verifying if the results are free of any p-hacking practices.
Author’s Response# Thank you for your suggestion. In response to your comment, we have added the degrees of freedom (df) to old Table 3 (new Table 4).
Comment 8# Results: Additionally, I highly recommend that the authors calculate the omega-squared effect size measure, which is less biased (the authors can refer to multiple publications by Daniel Lakens on that matter).
Author’s Response# Thank you for your valuable suggestion. In response to your comment, we have calculated the omega-squared (ω²) effect size measure instead of Partial Eta Squared and added it to old Table 3 (new Table 4).
Table 4. Mean difference in latent classes in depression, anxiety, life satisfaction, emotional problem, conduct problem, hyperactivity, peer problem, pro-social behaviors.
|
|
No risk M (SD) |
At risk M (SD) |
High risk M (SD) |
df |
F-value |
Sig. |
ω² |
|
Depression |
7.12 (7.49) |
11.83 (10.28) |
15.98 (14.39) |
2, 1003 |
45.730 |
<.001 |
0.082 |
|
Anxiety |
24.55 (6.37) |
28.33 (9.48) |
30.72 (10.49) |
34.792 |
<.001 |
0.063 |
|
|
Life satisfaction |
29.18 (6.37) |
25.96 (7.32) |
26.23 (8.00) |
19.965 |
<.001 |
0.036 |
|
|
Emotional problem |
2.52 (2.30) |
3.37 (2.53) |
4.15 (2.72) |
19.885 |
<.001 |
0.036 |
|
|
Conduct problem |
2.46 (2.16) |
3.31 (2.24) |
3.35 (2.18) |
14.058 |
<.001 |
0.025 |
|
|
Hyperactivity |
2.89 (2.29) |
3.73 (2.30) |
4.15 (2.43) |
15.468 |
<.001 |
0.028 |
|
|
Peer problem |
3.10 (2.24) |
3.67 (2.21) |
4.02 (2.44) |
8.146 |
<.001 |
0.014 |
|
|
Pro-social behaviors |
7.27 (2.40) |
6.69 (2.43) |
7.32 (2.41) |
4.072 |
.017 |
0.006 |
Note. Sig.= Significance; df= Degree of Freedom; ω²=Omega-Squared Effect Size
Comment 9# Results: in Table 3, eight effect size measure values are reported. Out of those 8 partial eta-squared values, 2 can be interpreted as small, 3 can be interpreted as moderate, and only 2 can be interpreted as large (above .05). The authors do not discuss the effect size values in both the Results section as well as the Discussion. It’s crucial to describe both the effect size measures and significance coefficients (such as p-values and confidence intervals). There is a difference between saying that two groups are different and stating that those groups are significantly different, but the size of the difference is very small.
Author’s Response# Thank you for your insightful comment. In response, we have now included the omega-squared (ω²) effect size measures in both the Results and Discussion sections.
Result section:
Results demonstrated different significance level among No-risk, At-risk, and High-risk classes for the key study variables. Depression (F = 45.730, p < .001, ω² = 0.082) and anxiety (F = 34.792, p < .001, ω² = 0.063) demonstrated medium effects, suggesting a moderate to strong impact of problematic QQ use. Life satisfaction (F = 19.965, p = .001, ω² = 0.036) and emotional symptoms (F = 19.885, p = .001, ω² = 0.036) have small to medium effects, and conduct problems (F = 14.058, p < .001, ω² = 0.025), hyperactivity (F = 15.468, p < .001, ω² = 0.028), and peer relationship problems (F = 8.146, p = .001, ω² = 0.014) demonstrated small effects. Additionally, prosocial behaviors demonstrated very small effect (F = 4.072, p = .017, ω² = 0.006), indicating least effect from problematic QQ use.
Discussion Section:
This cross-sectional study examined the associations between problematic QQ use and different mental health symptomatology among Chinese children and adolescents using Latent Class Analysis (LCA). The findings of the study highlighted a significant association between problematic QQ use and QQ addiction with mental health outcomes, e.g., depression, anxiety, emotional problems, conduct issues, hyperactivity, peer problems, and reduced life satisfaction. Comprehensive LCA identified three distinct classes of problematic QQ users: no-risk (77.2%), at-risk (16.8%), and high-risk (6.0%); with the comparatively higher risk groups demonstrated elevated addiction symptoms and negative mental health outcomes. Till date, this is the first study that explored the latent group of QQ users based the problematic QQ use symptoms. This findings are consistent with other studies those explored the latent group of user based on different other problematic social media platforms use, including Facebook and overall social media [34-37]. These findings align with existing literature on behavioral addiction and its adverse mental health impacts, highlighting the importance of early diagnosis and tailored intervention plans for individuals categorized in risk groups for developing behavioral addiction.
The findings derived from the LCA highlighted comorbidity of the all six components of addiction, along with higher negative mental health outcomes (depression, anxiety, Life satisfaction, emotional problem, conduct problem, hyperactivity, and peer problem) with significant mean differences. However, differences in mental health symptomatology were not same across the groups. Larger differences existed in depression and anxiety, while small to moderate differences in life satisfaction and emotional symptoms. Small differences were in conduct problems, hyperactivity, and peer relationship problems. Although differences were varied, these suggested vulnerability of high-risk group of QQ users.
Findings further showed significant associations between latent groups of QQ users and mental health symptomatology. High-risk group had a significant relationship with depression compared to no-risk group, which is in-line with previous studies, showing the association between excessive internet use and depressive symptoms among the adolescents [38-42]. It is highly anticipated that the problematic QQ use contributes to the elevated depressive symptoms through inducing social isolation, disrupted sleep quality, and confining other offline activities, e.g., fostering hobbies and outdoor pleasure activities which all ultimately result in negative mental health. . In contrast to the association with depression, Similar to the depressive symptoms, at-risk groups had a significant association with higher level of anxiety compared to no-risk group. This finding suggested different level of mental health vulnerability across the groups of QQ users.. Similar to the depressive symptoms, significant level of anxiety was also observed among the high-risk class than the no-risk class. The positive association between problematic social media use and higher anxiety level prevails among the adolescents whose primary source of socialization is based on virtual presence [45-48]. To explain this association, several contributing factors may surface, e.g., urge to maintain online presence, fear of missing out, social desirability standard. Often the validation seeking behaviors (for example: like, share, comment, and other interactions) lead to the emotional distress where the adolescents would be more vulnerable [49-52]. Cultural context plays a significant role in shaping the addictive behaviors of Chinese adolescents with regard to QQ use. Parenting style and academic stressors often play key roles in the development of problematic internet use in China [53]. In China, QQ is used not only to cope with academic stress but also for socialization, which can contribute to excessive and problematic use. Additionally, the multifaceted use of QQ (e.g., entertainment, social interactions, academic activities, etc.) makes it highly popular among adolescents.
Similar to anxiety, Our the findings also demonstrated a significant relationship between at-risk users and significantly lower levels of life satisfaction among the high-risk class compared to the no-risk users, class which is supported by the previous studies highlighting that, problematic internet use has multifaceted negative impacts on life satisfaction [54-57]. One of the plausible reasons for lower level of life satisfaction is due to the excessive QQ use which reduces the real-life physical activities and relationships. Adolescents those who prioritize their virtual presence over the physical interactions often result in loneliness and isolation and that jeopardizes the overall well-being [58-62].
Although present study failed to explore any significant associations between latent groups of QQ users and emotional problems, conduct problem, hyperactivity, peer problems, and prosocial behaviors, significant mean differences in this mental health symptomatology suggest possible vulnerability among at-risk and high-risk users. However, longitudinal and experimental studies (identifying at-risk and high-risk users, introducing interventions to reduce problematic usage symptoms, and examine the results) are needed to confirm these associations. Another important finding of the present study is related to effect of sex on the associations between latent groups of QQ users and mental health. Although existing literature suggests that females are more susceptible to negative mental health outcomes due to excessive internet use [43, 44], no moderating effect of sex on the key study variables was found in this study. This suggests that sex did not influence the development of negative mental health outcomes due to problematic QQ use.
Comment 10# Results: I believe that Table 4, representing the post-hoc analysis, should also report the effect size measure, such as Cohen’s d values.
Author’s Response# In response, we added the Cohen’s d values in old Table 4 (new Table 5).
Comment 11# Results: There is a small error in Table 5, since there is no such thing as an unstandardised beta coefficient. Did the authors mean to report the non-standardized OLS regression coefficient? Unstandardized regression coefficient should be reported as B (a capital letter b), while the standardized one is a Beta (represented by the Greek letter β).
Author’s Response# Thank you for addressing this issue. We corrected the typing mistake in the revision.
Comment 12# Results: I believe that the authors should report model specifications for the regression model: F-value, degrees of freedom, p-value, next to the R-squared values under the table.
Author’s Response# Thank you for this insightful comment. We added the model summary under each key variable.
|
Predictor |
B |
SE |
Β |
p-value |
95% CI |
|
|
Lower |
Upper |
|||||
|
Depression (Model Summary: F= 21.924, df (5, 1000), p < .001, R²= .314) |
||||||
|
No risk vs. At risk |
.028 |
.012 |
1.028 |
.024 |
1.004 |
1.054 |
|
No risk vs. High risk |
.061 |
.017 |
1.063 |
<.001 |
1.027 |
1.099 |
|
Anxiety (Model Summary: F= 16.163, df (5, 1000), p < .001, R²= .273) |
||||||
|
No risk vs. At risk |
.034 |
.013 |
1.034 |
.008 |
1.009 |
1.060 |
|
No risk vs. High risk |
.033 |
.018 |
1.034 |
.070 |
.997 |
1.071 |
|
Life satisfaction (Model Summary: F= 8.250, df (5, 1000), p < .001, R²= .199) |
||||||
|
No risk vs. At risk |
-.038 |
.013 |
.963 |
.004 |
.939 |
.988 |
|
No risk vs. High risk |
-.015 |
.021 |
.986 |
.490 |
.946 |
1.027 |
|
Emotional problem (Model Summary: F= 10.292, df (5, 1000), p < .001, R²= .199) |
||||||
|
No risk vs. At risk |
.012 |
.048 |
.988 |
.797 |
.899 |
1.085 |
|
No risk vs. High risk |
.025 |
.075 |
1.025 |
.740 |
.885 |
1.187 |
|
Conduct problem (Model Summary: F= 6.937, df (5, 1000), p < .001, R²= .183) |
||||||
|
No risk vs. At risk |
.084 |
.052 |
1.088 |
.103 |
.983 |
1.204 |
|
No risk vs. High risk |
.025 |
.083 |
1.025 |
.762 |
.872 |
1.206 |
|
Hyperactivity (Model Summary: F= 7.282, df (5, 1000), p < .001, R²= 187) |
||||||
|
No risk vs. At risk |
.029 |
.046 |
1.029 |
.528 |
.941 |
1.126 |
|
No risk vs. High risk |
.083 |
.069 |
1.086 |
.228 |
.950 |
1.243 |
|
Peer problem (Model Summary: F= 3.868, df (5, 1000), p= .002, R²= .138) |
||||||
|
No risk vs. At risk |
-.009 |
.048 |
.991 |
.856 |
.902 |
1.090 |
|
No risk vs. High risk |
.034 |
.075 |
1.034 |
.655 |
.892 |
1.199 |
|
Pro-social behaviors (Model Summary: F= 3.598, df (5, 1000), p =.003, R²= .133) |
||||||
|
No risk vs. At risk |
-.029 |
.039 |
.971 |
.454 |
.899 |
1.049 |
|
No risk vs. High risk |
.105 |
.067 |
1.110 |
.117 |
.974 |
1.266 |
Comment 13# Discussion: I highly encourage the authors to supplement this section with the size of the effect size measures in mind, as mentioned in my previous comment.
Author’s Response# Thank you for your insightful comment. In response, we have now included the omega-squared (ω²) effect size measures in the Discussion section.
Reviewer 2 Report
Comments and Suggestions for Authors
The study investigates the association between problematic use of the Chinese social media platform QQ and various mental health outcomes among 1,006 Chinese adolescents aged 11–17. Using Latent Class Analysis on the Problematic QQ Use Scale, the authors identified three subgroups:
- No-risk (77.2%)
- At-risk (16.8%)
- High-risk (6.0%)
Compared to the other groups, the high-risk group showed significantly higher levels of depression, anxiety, emotional and conduct problems, hyperactivity, and peer problems, as well as lower life satisfaction. Regression models confirmed that problematic QQ use predicted higher depression and anxiety scores, and lower life satisfaction (for the at-risk class). The study concludes that excessive QQ use is linked to poorer mental health among Chinese youth and calls for preventive and intervention strategies targeting high-risk users.
The paper addressed a timely and important topic. The design does have several limitations (listed below), but I think the paper can be published after a moderate revision.
Comments:
- The study situates itself within the growing literature on social media addiction and mental health. However, the theoretical contribution is limited. The authors reiterate well-established associations between excessive online engagement and psychopathology without proposing or testing novel mechanisms (e.g., self-regulation deficits, fear of missing out, social comparison). The introduction could be strengthened by a more explicit theoretical model linking problematic QQ use to specific emotional and behavioral outcomes. The following consensus paper could be a useful reference for this part: https://osf.io/preprints/psyarxiv/b94dy_v1
- The authors provide detailed model-fit information and post-hoc analyses. However, the regression section is confusingly reported—mixing odds ratios, unstandardized betas, and β coefficients inconsistently. Moreover, the effect sizes are small (partial η² mostly < .08), suggesting modest practical significance despite statistical robustness. The discussion overstates the strength of associations and fails to contextualize them against baseline mental health rates in Chinese adolescents.
- The design itself is fundamentally based on self-report measures. As also reported in the consensus statement above, self-reported measures have several limitations. Although the paper already discusses this point, I would like the authors to stress it further.
- The authors’ conclusions, calling for preventive programs, are reasonable but generic. There is little discussion of cultural or contextual factors, such as parental monitoring, academic stress, or the social functions of QQ in China. Furthermore, the paper does not address whether QQ addiction is qualitatively distinct from general social media overuse. A comparative perspective (e.g., QQ vs. WeChat or Douyin) would add depth.
- The manuscript is overall understandable but linguistically uneven and sometimes redundant. Numerous grammatical errors, awkward phrasing, and repeated statements (e.g., “homogenous with existing literature”) reduce clarity. The abstract is particularly verbose and could be condensed to highlight the novelty and main results more clearly.
- Also, I suggest the authors to avoid using acronyms in the title.
Author Response
The study investigates the association between problematic use of the Chinese social media platform QQ and various mental health outcomes among 1,006 Chinese adolescents aged 11–17. Using Latent Class Analysis on the Problematic QQ Use Scale, the authors identified three subgroups:
- No-risk (77.2%)
- At-risk (16.8%)
- High-risk (6.0%)
Compared to the other groups, the high-risk group showed significantly higher levels of depression, anxiety, emotional and conduct problems, hyperactivity, and peer problems, as well as lower life satisfaction. Regression models confirmed that problematic QQ use predicted higher depression and anxiety scores, and lower life satisfaction (for the at-risk class). The study concludes that excessive QQ use is linked to poorer mental health among Chinese youth and calls for preventive and intervention strategies targeting high-risk users.
The paper addressed a timely and important topic. The design does have several limitations (listed below), but I think the paper can be published after a moderate revision.
Comments:
Comment 1# The study situates itself within the growing literature on social media addiction and mental health. However, the theoretical contribution is limited. The authors reiterate well-established associations between excessive online engagement and psychopathology without proposing or testing novel mechanisms (e.g., self-regulation deficits, fear of missing out, social comparison). The introduction could be strengthened by a more explicit theoretical model linking problematic QQ use to specific emotional and behavioral outcomes. The following consensus paper could be a useful reference for this part: https://osf.io/preprints/psyarxiv/b94dy_v1
Author’s Response# We sincerely thank for you comment and mentioning the literature. We disagreed with your claim related to limited theoretical contribution and reiterate well-established associations between excessive online engagement and psychopathology without proposing. Most of the previous studies that you want to refer to have used variable-centered approach to explore the associations. In a variable-centered approach, there is no chance to examine the individual differences or group (based on latent characteristics) differences in these associations. In contrast, we used a person-centered approach through Latent Class Analysis (LCA), which allowed us to differentiate the individual difference based on the distinct pattern of problematic QQ use and its’ mental health implication. This approach enhanced the comprehension by identifying subclasses and their unique impact on mental health and thus offered a valuable novel insight on problematic QQ use and its effect on mental health. We strongly believe that this person-centered approach extended the literature considering the heterogeneity within the same population. Additionally, we would like to express the well align of the current introduction with the rationale. We mentioned that briefly in the last paragraph of the ‘Introduction’ section.
Comment 2# The authors provide detailed model-fit information and post-hoc analyses. However, the regression section is confusingly reported—mixing odds ratios, unstandardized betas, and β coefficients inconsistently. Moreover, the effect sizes are small (partial η² mostly < .08), suggesting modest practical significance despite statistical robustness. The discussion overstates the strength of associations and fails to contextualize them against baseline mental health rates in Chinese adolescents.
Author’s Response# Thank you for the comment. We updated the overall section in Table 6. We have updated the Discussion section accordingly.
Findings further showed significant associations between latent groups of QQ users and mental health symptomatology. High-risk group had a significant relationship with depression compared to no-risk group, which is in-line with previous studies, showing the association between excessive internet use and depressive symptoms among the adolescents [38-42].
Comment 3# The design itself is fundamentally based on self-report measures. As also reported in the consensus statement above, self-reported measures have several limitations. Although the paper already discusses this point, I would like the authors to stress it further.
Author’s Response# Thank you for the comment. We stressed the recall bias of self-reporting data, the updated section as follows:
This present study acknowledges some limitations. First, the self-reporting data are often prone to social desirability and recall bias. Respondents often answer in favorable state that distorts the information which may lead to the inaccurate findings.
Comment 4# The authors’ conclusions, calling for preventive programs, are reasonable but generic. There is little discussion of cultural or contextual factors, such as parental monitoring, academic stress, or the social functions of QQ in China. Furthermore, the paper does not address whether QQ addiction is qualitatively distinct from general social media overuse. A comparative perspective (e.g., QQ vs. WeChat or Douyin) would add depth.
Author’s Response# Thank you for the comment. We extended our Discussion
Cultural context plays a significant role in shaping the addictive behaviors of Chinese adolescents with regard to QQ use. Parenting style and academic stressors often play key roles in the development of problematic internet use in China [51]. In China, QQ is used not only to cope with academic stress but also for socialization, which can contribute to excessive and problematic use. Additionally, the multifaceted use of QQ (e.g., entertainment, social interactions, academic activities, etc.) makes it highly popular among adolescents.
And Conclusion:
This present study explores the association between problematic QQ use and mental health outcomes among Chinese adolescents through latent classification. The study demonstrated that the high-risk class reported significantly higher depression, anxiety, emotional problems, conduct issues, along with lower levels of life satisfaction compared to the no-risk and at-risk classes. It is necessary to consider the unique cultural and social impact of problematic QQ use in China where parental style, academic stress, and social interaction are important. These findings affirm the importance of early diagnosis and intervention for children and adolescents, who are at risk of problematic QQ use, especially those classified as at-risk and high-risk individuals. The findings of this present study highlight the mental health tolls of problematic QQ use, which encourages mental health professionals and educators to develop reasonable intervention strategies emphasizing the personality and individual differences of minors in digital settings. Effective intervention program should aim to consider the cultural norms to promote the healthy digital presence of the adolescents.
Comment 5# The manuscript is overall understandable but linguistically uneven and sometimes redundant. Numerous grammatical errors, awkward phrasing, and repeated statements (e.g., “homogenous with existing literature”) reduce clarity. The abstract is particularly verbose and could be condensed to highlight the novelty and main results more clearly.
Author’s Response# Thank you for your comment. We checked the manuscript and made necessary corrections.
Revised Abstract:
The rise of problematic social media use among children and adolescents is often associated with significant physical and psychosocial effects. In China, QQ, a popular social media platform among youth, has become a major mental health concern due to its excessive use. This present study aimed to explore the association between QQ addiction and mental health outcomes through Latent Class Analysis (LCA). The study data were collected from a sample of 1,006 Chinese school students (49.8% male; age M = 13.32, SD = 1.34 years) through a paper-pencil survey using the convenience sampling technique. LCA identified three latent groups based on QQ addiction symptoms scores: no-risk (77.2%), at-risk (16.8%), and high-risk (6.0%). The analysis revealed that children and adolescents in the high-risk class exhibited significantly higher levels of depression, anxiety, emotional problems, conduct issues, hyperactivity, and peer problems, as well as lower life satisfaction and pro-social behaviors compared to the no-risk and at-risk groups (p < 0.05), signifying a strong association between problematic QQ use and poor mental health. Mental health professionals would benefit from designing intervention plans to mitigate the negative mental health outcomes among the high risk and at risk classes of problematic QQ users.
Comment 6: Also, I suggest the authors to avoid using acronyms in the title.
Author’s Response# Thank you for your comment. We used ‘QQ’ in the title as there is no other name or extended name of this social media platform.
Reviewer 3 Report
Comments and Suggestions for Authors
Dear Authors,
There are some suggestions:
- The Cronbach's alpha is missing for all instruments you used, as well as the descriptive statistics for total score. It would be the best solution to add the new table with this data.
- In the Discussion part, comparison with the results of other researchers could be more detailed.
- It would be very useful to compare all results according to the sex of the respondents. It is very important since almost all papers indicate that misuse of the internet and social networks is determined by sex. Also, many researches of depression and anxiety point out that female respondents are more susceptible to this. So sex could be an important moderator variable.
- The conclusion part should be longer and more consistent.
Author Response
Dear Authors,
There are some suggestions:
Comment 1# The Cronbach's alpha is missing for all instruments you used, as well as the descriptive statistics for total score. It would be the best solution to add the new table with this data.
Author’s Response: Thank you for pointing out this inconsistency, we now reported the Cronbach's alpha for all the scales.
In this present study, the PQQUS demonstrated good internal consistency with a Cronbach’s alpha of 0.857.
In this present study, the BAI demonstrated good internal consistency with a Cronbach’s alpha of 0.927. .
In this present study, the BDI demonstrated good internal consistency with a Cronbach’s alpha of 0.907.
In this present study, the SWLS demonstrated good internal consistency with a Cronbach’s alpha of 0.883.
In this present study, the overall SDQ demonstrated good internal consistency with a Cronbach’s alpha of 0.774 (emotional problems: 0.694, conduct problem: 0.677, hyperactivity: 0.675, peer problems: 0.709, prosocial behaviors: 0.698).
Comment 2# In the Discussion part, comparison with the results of other researchers could be more detailed.
Author’s Response: Thank you for the comment. We added comparison with previous studies and the cultural context in the revised Discussion section.
Comment 3: It would be very useful to compare all results according to the sex of the respondents. It is very important since almost all papers indicate that misuse of the internet and social networks is determined by sex. Also, many researches of depression and anxiety point out that female respondents are more susceptible to this. So sex could be an important moderator variable.
Author’s Response# Thank you for the comment. In response, we conducted the moderation effect of sex on the relationship between problematic QQ use and mental health outcomes. However, no significant moderation effect of sex on the key outcomes was found (see the Supplementary Figure 1).
Comment 4: The conclusion part should be longer and more consistent.
Author’s Response# Thank you for the comment. We extended or conclusion in the revised manuscript now.
This present study explores the association between problematic QQ use and mental health outcomes among Chinese adolescents through latent classification. The study demonstrated that the high-risk class reported significantly higher depression, anxiety, emotional problems, conduct issues, along with lower levels of life satisfaction compared to the no-risk and at-risk classes. It is necessary to consider the unique cultural and social impact of problematic QQ use in China where parental style, academic stress, and social interaction are important. These findings affirm the importance of early diagnosis and intervention for children and adolescents, who are at risk of problematic QQ use, especially those classified as at-risk and high-risk individuals. The findings of this present study highlight the mental health tolls of problematic QQ use, which encourages mental health professionals and educators to develop reasonable intervention strategies emphasizing the personality and individual differences of minors in digital settings. Effective intervention program should aim to consider the cultural norms to promote the healthy digital presence of the adolescents.
Reference
Andreassen, C. S., Billieux, J., Griffiths, M. D., Kuss, D. J., Demetrovics, Z., Mazzoni, E., & Pallesen, S. (2016). The relationship between addictive use of social media and video games and symptoms of psychiatric disorders: A large-scale cross-sectional study. Psychology of Addictive Behaviors, 30(2), 252-262. https://doi.org/10.1037/adb0000160
Gu, H., Shi, B., He, H., Yuan, S., Cai, J., Chen, X., & Wan, Z. (2025). Association Between Excessive Internet Use Time, Internet Addiction, and Physical-Mental Multimorbidity Among Chinese Adolescents: Cross-Sectional Study. Journal of medical Internet research, 27, e69210. https://doi.org/10.2196/69210
Li, J., Sun, W., Luo, Z., Liu, Y., Huang, X., Jiang, D., Li, S., Meng, J., Gu, F., Zhang, R., & Song, P. (2024). Dose-Response Associations of Internet Use Time and Internet Addiction With Depressive Symptoms Among Chinese Children and Adolescents: Cross-Sectional Study. JMIR public health and surveillance, 10, e53101. https://doi.org/10.2196/53101
Soriano-Molina, E., Limiñana-Gras, R., Patró-Hernández, R., & Rubio-Aparicio, M. (2025). The association between internet addiction and adolescents’ mental health: A Meta-Analytic Review. Behavioral Sciences, 15(2), 116. https://doi.org/10.3390/bs15020116
Ye, X.-L., Zhang, W., & Zhao, F.-F. (2023). Depression and internet addiction among adolescents:a meta-analysis. Psychiatry Research, 326, 115311. https://doi.org/10.1016/j.psychres.2023.115311
Zhang, J., Wang, E., Zhang, L., & Chi, X. (2024). Internet addiction and depressive symptoms in adolescents: Joint trajectories and predictors. Frontiers in Public Health, 12, 1374762. https://doi.org/10.3389/fpubh.2024.1374762
Round 2
Reviewer 3 Report
Comments and Suggestions for Authors
Dear Authors,
Althought I am aware that might be suggestion of other reviewer, I strongly advise you to return your explanations in the Discussion part - page 14 (lines 437-460). I do find this is very appropriate and relevant.
Author Response
Comment 1: The English is fine and does not require any improvement.
Author’s Response# Thank you for addressing this issue. One English native language expert revised the overall manuscript and we are now confident about the Brain Sciences standard.
Also, we updated the Figure 1 with high quality PNG format.
Comment 2: Although I am aware that might be suggestion of other reviewer, I strongly advise you to return your explanations in the Discussion part - page 14 (lines 437-460). I do find this is very appropriate and relevant.
Author’s Response# Thank you for your valuable comment. We now added the previous explanation in ‘Discussion’ as per you comment.
Thank you for your time to review our manuscript to disseminate the scientific message more vividly for the reader.